# A Nonrandomized Phase 2 Trial of EG-Mirotin, a Novel, First-in-Class, Subcutaneously Deliverable Peptide Drug for Nonproliferative Diabetic Retinopathy

**DOI:** 10.3390/medicina59010178

**Published:** 2023-01-16

**Authors:** Seunghoon Yoo, Dae Hyuk You, Jeongyoon Lee, H. Christian Hong, Sung Jin Lee

**Affiliations:** 1Nune Eye Hospital, Seoul 06198, Republic of Korea; 2Department of Statistics, College of Letters & Science, University of Wisconsin-Madison, Madison, WI 53715, USA; 3Brain and Cognitive Sciences Research Center, Daegu Gyeongbuk Institute of Science and Technology (DGIST), Daegu 42988, Republic of Korea; 4Department of Medicine and Medical Microbiology, School of Medicine, New Tokyo Medical College, Pohnpei FM 96941, Micronesia; 5Department of Ophthalmology, Soonchunhyang University College of Medicine, Soonchunhyang University Hospital, Seoul 04401, Republic of Korea

**Keywords:** diabetic retinopathy, nonproliferative diabetic retinopathy, retinal ischemia

## Abstract

*Background and objectives:* EG-Mirotin (active ingredient EGT022) targets nonproliferative diabetic retinopathy (NPDR), the early stage of retinopathy. EG-Mirotin reverses capillary damage before NPDR progresses to an irreversible stage. EG-Mirotin safety and efficacy were investigated in patients with type 1 or type 2 diabetes mellitus and moderate to severe NPDR. *Methods:* In this open-label, single-arm, single-center, exploratory phase II study, 10 patients (20 eyes) received EG-Mirotin once a day (3 mg/1.5 mL sterile saline) for 5 days and were evaluated for ischemic index changes and safety. End of study was approximately 8 ± 1 weeks (57 ± 7 days) after the first drug administration. *Results:* EG-Mirotin injections were well tolerated, with no dose-limiting adverse events, serious adverse events, or deaths. Four treatment-emergent adverse events (TEAEs) unrelated to the investigational drug were observed in 2 out of 10 participants (20%) who had received the investigational drug. The overall average percent change in ischemic index at each evaluation point compared with baseline was statistically significant (Greenhouse–Geisser F = 9.456, *p* = 0.004 for the main effect of time), and a larger change was observed when the baseline ischemic index value was high (Greenhouse–Geisser F = 10.946, *p* = 0.002 for time × group interaction). *Conclusions:* The EG-Mirotin regimen established in this study was shown to be feasible and safe and was associated with a trend toward potential improvement in diabetes-induced ischemia and retinal capillary leakage.

## 1. Introduction

Even in the 21st century, diabetes mellitus is rapidly spreading as a global epidemic. Currently, 463 million adults (20–79 years) are affected worldwide, and the number will reach 700 million by 2045 [1]. Diabetic retinopathy (DR), a disease complication, is the leading cause of blindness in people aged 20–74 years [2]. DR is caused by diabetes-induced injury of the retinal microvasculature, and it is classified as nonproliferative DR (NPDR) or proliferative DR (PDR) [3]. Furthermore, diabetic macular edema (DME) can cause severe visual impairment [4].

Diabetes damages endothelial cells [5], which stimulates cell proliferation and motility. Microvessels are formed by vascular endothelial cells and pericytes that support and surround them [6,7,8]. Pericytes contribute to the stabilization of capillary beds, and their correct interaction with the extracellular matrix is essential for microvascular development [9]. If diabetes-induced changes worsen, pericyte loss and basement membrane thickening will eventually lead to the loss of microvessels [10]. These vascular abnormalities may lead to inadequate blood flow in retinal areas. To adjust for hypoperfusion, angiogenesis occurs, causing undesired neovascularization, retinal detachment, and, eventually, the loss of sight [11]. In patients with diabetes mellitus for more than 20 years, an estimated 86% and 77% of patients with type 1 and type 2 diabetes, respectively, experience loss of vision [12]. Moreover, patients with DR contribute disproportionately to healthcare and societal costs caused by diabetes mellitus.

DR prevention consists primarily of the optimal control of blood glucose levels [13], as well as the treatment of other risk factors, especially hypertension and hyperlipidemia. However, no appropriate therapies for early-stage retinopathy (NPDR) and only limited treatment options for late-stage retinopathy (PDR) exist. Once apparent, PDR can be treated by laser coagulation and intraocular injection of antiangiogenic drugs, such as bevacizumab, ranibizumab, and pegaptanib [14,15]. Current DR treatment involves injecting antivascular endothelial growth factor (anti-VEGF) antibodies to inhibit angiogenesis in the ischemic retina and prevent vascular leakage caused by pericyte loss [16]. However, this treatment cannot restore the retina because capillaries are irreversibly damaged. Therefore, to improve outcomes, new treatment paradigms are required. It is necessary to delay vision loss as much as possible by preventing damage to capillaries or restoring them in the NPDR stage.

EyeGene Inc. developed the new drug EG-Mirotin, which contains the active ingredient EGT022, obtained from a human-derived protein called ADAM15 [8]. EG-Mirotin targets NPDR before capillary damage progresses to an irreversible stage and offers a completely new treatment approach. Most patients with DR experience anxiety when they are informed that the drug is injected into their eyes. However, EG-Mirotin is a novel product that can alleviate patients’ anxiety, as it can be injected subcutaneously with a small gauge (31–29) needle.

After the subcutaneous injection, EGT022 binds to integrins on the surface of thrombocytes, stimulating their production of growth factors including angiopoietin 1. In DR, capillaries are prone to rupture and subsequent retinal hemorrhages and fluid leakages, resulting in retinal and macular edemas [17]. The primary activity of thrombocyte-released growth factors, especially angiopoietin 1, following EGT022 administration appears to be the recruitment and development of pericytes, specialized connective tissue cells, which, together with endothelial cells, constitute capillaries [18].

EG-Mirotin has been shown to be safe in a phase 1 clinical trial (unpublished data). However, the efficacy of EG-Mirotin has not been clarified. The current trial aimed to evaluate the efficacy and safety of subcutaneous EG-Mirotin injection in adult NPDR patients with diabetic macular edema.

## 2. Materials and Methods

### 2.1. Study Drug

EG-Mirotin was provided by EyeGene Inc. (Seoul, Korea) as a freeze-dried powder. EGT022 was cloned using the N-terminal primer 5’-CTCGAGAAAGAACCTGCCAGCTGA-3’ and the C-terminal primer 5’-GCGGCCGCTCAGCCATCCCCTAGGC-3’ from the human fetal liver cDNA library. We used the previously described Pichia pastoris expression system [19].

### 2.2. Trial Design

This was an open-label, single-arm, single-center, exploratory phase II study. The protocol was approved by the Ministry of Food and Drug Administration of Korea (approval no. 32764) and by the investigational review board of Soonchunhyang University Hospital. Patients gave written consent after being informed about the characteristics of the clinical trial and the investigational drug.

### 2.3. Study Population

We enrolled 10 patients (8 men, 2 women) aged >19 years with moderate to severe NPDR (Table 1). The key inclusion criteria were: type 2 diabetic patients undergoing diabetes treatment, moderate to severe NPDR, and patients not under photocoagulation and anti-VEGF treatment. The key exclusion criteria were: presence of any eye disease or disorder other than DR that can affect the evaluation, vitrectomy or intravitreal injection of anti-VEGF drugs, a major ophthalmic surgery in the last 3 months, those who were under intensive insulin therapy, uncontrolled hypertension, kidney or liver failure, history of malignant cancer within 3 years, immunodeficiency or autoimmune diseases, history of hypersensitivity to EG-Mirotin components, pregnant or breastfeeding women, and patients who were under another clinical trial.

### 2.4. Drug Administration

Enrolled patients received EG-Mirotin once per day (3 mg in 1.5 mL of sterile saline) for 5 days with an additional follow-up period of approximately 7 weeks (8 weeks in total: screening (V1), treatment period (V2–V6), follow-up period (V7–V9)). End of study was approximately 8 ± 1 weeks (57 ± 7 days) after the first dose.

### 2.5. Evaluation of the Ischemic Index and Leakage Index

Ischemic index and leakage index values were measured using ultrawide-field fluorescein angiography (UWFA) (Figure 1) with Optos California (Optos PLC, Dunfermline, UK). To determine the indexes, total retinal area, ischemic area, and leakage area were measured using the Optos Advance software (Optos PLC). On two UWFA images obtained at approximately 40 s and 8 min after contrast medium injection, an evaluator marked the maximal identifiable retina with a continuous line to calculate total retinal areas (in mm^2^). Summed retinal areas were marked by the assessor as nonperfused regions in the first image, while the leaking capillaries in the second image were defined as ischemic and leakage areas. The ischemic and leakage indexes (in %) were defined as the ratios of ischemic and leakage areas divided by the total retinal area.

During the treatment period (V2–V6), the participants were monitored for the safety and efficacy of the treatment. The ischemic and leakage indexes were first measured at baseline (V1) and subsequently at 2, 4, and 8 weeks after administration (V7, V8, and V9, respectively).

### 2.6. Evaluation Criteria

The primary endpoint was the mean percent change in ischemic index at each assessment point compared with baseline. The secondary endpoints were: (i) fraction of the participants who progressed to PDR, (ii) DME-related changes compared with baseline, (iii) changes in the thickness of cerebrospinal fluid (CSF) and the retinal volume, (iv) visual acuity, (v) improvement and deterioration of the eye compared with baseline, (vi) vascular leakage, and (vii) changes in severity and frequency of peripheral lesions compared with baseline measured at each assessment time point.

Safety endpoints included: (i) incidence and characteristics of adverse events after study drug administration; (ii) incidence and characteristics of ocular adverse events; and (iii) physical examination, vital signs, electrocardiogram, and clinical laboratory examination findings related to the administration of the investigational drug.

### 2.7. Statistical Analysis

The study population was divided into a safety assessment group, referring to the patients undergoing safety-related follow-up, and a full assessment group, representing the patients with whom the drug efficacy was tested. Statistical analysis of the primary endpoint of this study was performed as a two-tailed test with a significance level (α) of 0.1; all other analyses were based on two-tailed tests with an α of 0.05.

Data were analyzed based on the number of participants, incidence rates, and number of adverse events related to the investigational drug, as well as their severity, measures taken, and outcomes. The number of participants by system organ class (SOC) and preferred term (PT), incidence rates, and number of occurrences were summarized. Descriptive statistics of the vital signs and quantitative laboratory tests represented the degree of change before and after the investigational drug administration. Changes within the administration groups were tested for differences compared with baseline using the paired *t*-test or Wilcoxon signed-rank test. Significance of changes within the drug administration group was statistically tested using McNemar’s or Bowker’s test.

To further analyze EG-Mirotin’s effects, the V1 cutoff indicating a significant decrease in ischemic index was determined using the ‘maxstat’ R package, and two groups were defined based on this cutoff. The repeated measures ANOVA test was used to identify changes in ischemic and leakage indexes over time. To assess the difference in average ischemic index between these two groups, the Wilcoxon test, repeated measures ANOVA, and Bonferroni post hoc analysis were used. R version 4.0.4 (R Project for Statistical Computing, Vienna, Austria), SPSS version 28 (IBM, New York, NY, USA), and SAS version 9.4 (SAS Institute, North Carolina, USA) were used for statistical analyses and graphical representation.

## 3. Results

### 3.1. Study Population

A total of 10 subjects were selected based on the inclusion and exclusion criteria described in the materials and methods section. The average interval between visits in the treatment period was 23.22 ± 3.32 days between baseline (V1) and V2, 12.12 ± 3.36 days between V2 and V3, and 31.89 ± 5.52 days between V3 and V4 (Figure 2). The follow-up period consisted of

### 3.2. Safety Evaluation

Four treatment-emergent adverse events (TEAEs) were observed in 2 out of 10 participants (20.00%) who received EG-Mirotin. None of those TEAEs were related to the investigational drug and did not result in drug discontinuation. No adverse events leading to death were reported.

### 3.3. Drug Efficacy

#### 3.3.1. Primary Efficacy Outcomes

The primary efficacy outcome, the overall average percent change in ischemic index at each evaluation point compared with baseline, was statistically significant (Greenhouse–Geisser F = 9.456, *p* = 0.004 for the main effect of time), and a larger change was observed when the baseline value was high (Greenhouse–Geisser F = 10.946, *p* = 0.002 for time × group interaction). The baseline (V1) ischemic index for the 20 enrolled eyes was 3.82 ± 4.78%, and the corresponding indexes at V7, V8, and V9 after study drug administration were 3.50 ± 4.66%, 3.59 ± 4.24%, and 3.11 ± 3.85%, respectively. The changes from baseline were −0.28 ± 0.46%, −0.52 ± 0.76%, and −0.71 ± 1.02% (V7: *p* = 0.0025, V8: *p* = 0.0095, and V9: *p* = 0.0005, respectively). When the baseline ischemic index was >1.59%, a statistically significant improvement over time was observed (>1.59 group: Wilk’s lambda F = 9.142, *p* = 0.002 for the simple main effect of time). By contrast, no improvement or deterioration was observed in patients with a low ischemic index value at baseline.

#### 3.3.2. Secondary Efficacy Outcomes

Progression to PDR (ETDRS severity ≥61) was not observed. Changes in DME degree (without AMD, nonclinically significant macular edema): (i) most changes were not observed from baseline; (ii) 2 of 6 eyes (33.33%) assessed as SMD at baseline improved to grade ‘no SMD’ after treatment (V8 and V9); (iii) none of the eyes had a worse edema degree; (iv) no significant changes in CSF thickness and retinal volume were observed between time points before and after drug administration; and (v) no significant changes in best-corrected ETDRS visual acuity were observed compared with baseline.

Changes in DRSS scores from baseline: (i) no eye improved 1 or 2 steps; (ii) one eye was categorized as deterioration of 1 or 2 steps; and (iii) no eye had ≥3 steps (improvement or deterioration).

Change in vascular leakage rate from baseline: The average leakage index decreased significantly over time (Greenhouse–Geisser F = 6.442, *p* = 0.008 for the overall main effect of time). After test drug administration, changes at V7, V8, and V9 were significantly different with −0.69 ± 1.17%, −0.85 ± 1.53%, and −2.09 ± 2.32%, respectively (V7: *p* = 0.0233, V8: *p* = 0.0305, and V9: *p* = 0.0001, respectively).

Changes in the severity and frequency of peripheral lesions: (i) retinal hemorrhage/retinal microvascularization: ‘no change’ in severity in 18 eyes (90.00%), ‘best’ in 2 eyes (10.00%), and no changes in the distribution; (ii) venous beading: ‘no change’ in severity compared with baseline for all V9 visits (100.00%), one eye each with dilated venous beading with intraretinal and superonasal microvascular abnormalities; and (iii) severity: ‘unchanged’ in 18 eyes (90.00%), ‘best’ in 2 eyes (10.00%), and no changes in the distribution.

### 3.4. Ischemic Index

The UWFA images of the eyes were assessed before (V1, baseline) and 2, 4, and 8 weeks after (V7, V8, and V9, respectively) drug administration. There was an overall significant decrease in ischemic indexes during the investigation period (Greenhouse–Geisser F = 9.456, *p* = 0.004 for the main effect of time). The average ischemic index decreased over time, except for the interval between V7 and V8 (Figure 3). EG-Mirotin administration significantly decreased the average ischemic index values from baseline (3.82 ± 4.78) to V9 (3.11 ± 3.85; *p* = 0.0005). The ischemic percent difference (IPD) was defined as the differences in ischemic indexes between V1 and V7 (IPD1, Figure 4A), V1 and V8 (IPD2, Figure 4B), and V1 and V9 (IPD3, Figure 4C). Patients were divided into two groups based on an ischemic index cutoff of 1.59% at baseline. The IPD and the ischemic index of all eyes with a cutoff value of >1.59 decreased over time (Figure 4 and Figure 5). The two groups were significantly different regarding IPD1 (*p* < 0.001), IPD2 (*p* = 0.004), and IPD3 (*p* < 0.001, Wilcoxon test; Figure 4).

To observe changes in ischemic indexes within each group, the repeated measures ANOVA test was used. The ischemic index significantly decreased over time (Greenhouse–Geisser F = 9.456 for the main effect of time), and a significant interaction effect was observed (Greenhouse–Geisser F = 10.946 for time × group interaction). A significant change in the ischemic index was found in the >1.59 group (Wilk’s lambda F = 9.142, *p* = 0.002 for the simple main effect of time), whereas no change was observed in the ≤1.59 group (Wilk’s lambda F = 0.038, *p* = 0.990 for the simple main effect of time).

In the >1.59 group, the ischemic index was significantly different compared with the baseline value at V7, V8, and V9 (*p* = 0.001, *p* < 0.001, and *p* = 0.001, respectively; Bonferroni post hoc test). For the ≤1.59 group, the ischemic index did not significantly differ among visits. In the >1.59 group, EG-Mirotin administration decreased the average ischemic index from 7.13 ± 4.86 at baseline to 5.71 ± 3.97 at V9 (Figure 5 and Figure 6). Thus, for patients with an ischemic index above the 1.59% cutoff, EG-Mirotin administration significantly decreased the ischemic index value.

### 3.5. Retinal Leakage Index

The UWFA images of the patients’ eyes were assessed before (V1, baseline) and 2, 4, and 8 weeks after (V7, V8, and V9, respectively) drug administration. There was an overall decrease in leakage indexes across the visits (Greenhouse–Geisser F = 6.442, *p* = 0.008 for the main effect of time). The average leakage index decreased over time, except for the interval between V7 and V8. Although the mean value increased between V7 and V8, this change was not statistically significant (*p* = 0.14). EG-Mirotin administration decreased the average leakage index from baseline (13.60 ± 11.21) to V9 (11.51 ± 9.80, *p* = 0.0001; Figure 7).

To identify whether ischemic and leakage indexes were correlated at each time point, Spearman’s rank correlation test was conducted, as these indexes were not normally distributed (Shapiro–Wilk w = 0.7868, *p* < 0.0001 for ischemic index and w = 0.9167, *p* < 0.0001 for leakage index). The two indexes were moderately correlated (Spearman’s ρ = 0.3779, *p* = 0.0007; Figure 8).

## 4. Discussion

Diabetes mellitus is one of the most persistent diseases in human history. Despite the tremendous achievements in diabetes research, this disease still negatively affects patients’ well-being with numerous complications in varying loci and severity grades [20]. Diabetic retinopathy (DR), a common but severe ocular complication, is a major cause of blindness and, thus, greatly diminishes the quality of life [21]. Currently, there is no approved drug for early-stage DR, i.e., nonproliferative diabetic retinopathy (NPDR), and patients are without treatment until the disease progresses to proliferative diabetic retinopathy (PDR) [22,23]. We believe that improving the quality of life of patients with NPDR by slowing or halting the progression of visual impairment with an early-stage intervention is a challenge in modern medicine.

Recent DR treatment trends have two main therapeutic goals: (i) prevention of progression from NPDR to PDR, which has a high risk of visual loss due to vitreous hemorrhage or tractional retinal detachment, and (ii) prevention of loss of central vision due to DME [24]. To achieve these goals, it is necessary to prevent capillary leakage or suppress neovascularization caused by increased VEGF levels in the eye due to retinal ischemia [25]. Panretinal photocoagulation (PRP), which has been used for a long time, effectively lowers VEGF levels. However, PRP is known to have side effects, particularly with reduced visual acuity due to the destructive nature of photocoagulation and decreased visual acuity thanks to increased macular edema [26]. A recently popular method, intraocular injection of anti-VEGF antibodies, is a potent therapy that can achieve these goals more rapidly [27]. However, this approach aims at short-term effects in an irreversible disease stage in which blood vessels cannot return to a normal state and the retinal condition continues to gradually deteriorate [28,29]. If it were possible to reduce retinal capillary damage and related retinal ischemia, which are the causes of capillary leakage and neovascularization, the risk of blindness could be reduced by preventing the progression of DME and PDR.

Diabetes is a disease in which high levels of circulating blood sugar damage blood vessels [30]. In the human body, the kidneys and retina are the organs with the highest number of capillaries and the highest blood flow per unit area. Therefore, these organs are mainly affected by diabetes, and retinal damage ultimately increases the risk of blindness [31]. Abnormally high blood sugar levels in diabetes can damage endothelial cells, which in capillaries consist of only one cell layer [32,33]. For about 5 years of diabetes, cells surrounding the capillaries restore this endothelial cell layer and ensure normal capillary function. Afterward, even these surrounding cells are damaged, and capillary microaneurysms form, causing vessel occlusions and hemorrhages [19]. Perivascular cells vanish, capillaries are irreversibly damaged, the retina cannot maintain its normal morphology, and DR eventually progresses regardless of treatment [30]. With the expansion of retinal ischemic areas, VEGF levels increase to stimulate the development of new retinal capillaries counteracting ischemia [30].Therefore, if it were possible to maintain or revive the surrounding cells at the stage of endothelial cell damage, retinal capillaries could be rescued, and ischemia could be reduced.

EG-Mirotin was developed to prevent retinal damage and blindness by restoring capillaries before their damage progresses to an irreversible stage. Existing anti-VEGF antibodies for intraocular injection reduce damage to new blood vessels in the irreversible stage [34,35]. Although aflibercept (Eylea), an anti-VEGF treatment, is approved for NPDR in the USA, the need for frequent ocular injections burdens patients and physicians, thus precluding accessibility at the NPDR stage [36]. EG-Mirotin was developed to prevent capillary damage at the earliest stage of NPDR so that the disease does not progress to an irreversible stage, with the additional advantage of subcutaneous administration. The findings of this study confirmed that EG-Mirotin administration restored the capillaries where retinal ischemia occurred, thereby reducing the ischemic area in the retina. By blocking irreversible capillary damage via the recruitment of surrounding cells and actively initiating treatment at the NPDR stage, progression to the proliferative stage can be largely prevented [37]. Since EG-Mirotin is subcutaneously administered instead of into or around the eye, the patient’s fear of ocular injection is reduced. Given the fact that the mechanism of this drug constitutes a novel approach, research data are continually accumulating. In the near future, more research is expected to be conducted on EG-Mirotin in order to develop better drugs for patients with NPDR.

In the present study, progression to proliferative DR did not occur in any of the participants. Moreover, both ischemic and leakage index values determined in UWFA images of lesions related to fundus microvascularization were significantly reduced compared with baseline values. Of note, a greater degree of change was observed in patients who had a larger ischemic index value at baseline. The participants were further categorized into two groups based on the cutoff of 1.59% in the baseline ischemic index. There was a significant interaction between group assignment and therapeutic effect over time, of which statistical significance was only observed in the >1.59 group in the post hoc comparison. It is expected that the ≤1.59 group showed no difference in the ischemic index over time because there was little room for change, as the initial ischemic value was ≤1.59%, with a few patients having an ischemic index of zero. The lack of index change can also be interpreted as the absence of disease deterioration. As retinal ischemia is a major factor in the pathogenesis of PDR [38], the prevention of ischemia may protect patients from progression to PDR. A larger placebo-controlled follow-up study with longer observation duration will be required to confirm the protective effects of EG-Mirotin. EGT022 is a protein containing an arginine–glycine–aspartic acid sequence [39]. Our former experimental data suggest that EGT022 restores retinal blood vessels in the OIR mouse model [9], while EGT022-induced maturation of functional retinal blood vessels is closely associated with pericyte recruitment and blood vessel integrity [8]. It can be assumed that the significant reduction in ischemia and leakage can be attributed to the active ingredient of EG-Mirotin and its molecular mode of action of pericyte recruitment [40], thereby stabilizing blood vessels.

We found no serious adverse effects or safety concerns following EG-Mirotin administration. None of the participants reported TEAEs that resulted in drug discontinuation. In preclinical studies, EG-Mirotin had no acute or genotoxicity. EG-Mirotin alone did not induce angiogenesis and had no growth-promoting effects on malignancies [41]. The efficacy of EG-Mirotin in retinal edema has been demonstrated in the Miles assay as a supplemental assay to assess the vascular permeability of microvessels. EG-Mirotin appeared to be effective in ameliorating retinopathy-related abnormalities for up to 3 weeks, as demonstrated in a streptozotocin-induced rat model of DR [9].

Despite showing promising results, this study has a few limitations. Based on our preclinical experience, we chose to use 3 mg of EG-Mirotin, which may not be a sufficient dose to cause substantial changes in retinal blood vessels. Theoretically, the dose could be increased to 10 mg per administration. However, we need to collect more safety data to justify a dose increase. The small number of subjects in this trial limits our ability to precisely define safety and efficacy. In the IIb clinical trial that we are planning for the future, we believe that between 100 and 300 participants should be enrolled, which is a generally accepted number of participants. Because type 1 and type 2 diabetes play different mechanisms, this study should be considered separately with sufficient samples. Hormonal changes in studies of gender differences should be considered in future trials. In the present study, the participants visited the clinic during the treatment period (i.e., a total of 5 days: V2–V6). A daily clinic visit is a nuisance for many patients. Therefore, our future goal is to prepare EG-Mirotin for self-injection, as established for insulin administration, so patients will not have to visit the clinic daily for their drug treatment.

## 5. Conclusions

We developed EG-Mirotin (EGT022), a novel, first-in-class, subcutaneously deliverable peptide drug for NPDR, for which currently no effective drug exists. The EG-Mirotin regimen established in this study was shown to be feasible and safe and was associated with a trend toward potential improvement in diabetes-induced ischemia and retinal capillary leakage. This trial found no serious adverse events due to the administration of EG-Mirotin. Moreover, the subcutaneous injection of this drug would remove patients’ fear and anxiety of being intraocularly injected. Therefore, EG-Mirotin would preserve the well-being of patients with DR at the reversible stage by greatly reducing the risk of loss of vision, with the added benefit of convenient drug administration. To verify the therapeutic potential of EG-Mirotin in DR, further comprehensive clinical trials focusing on patients with higher baseline ischemic indexes are recommended.

## Figures and Tables

**Figure 1 medicina-59-00178-f001:**
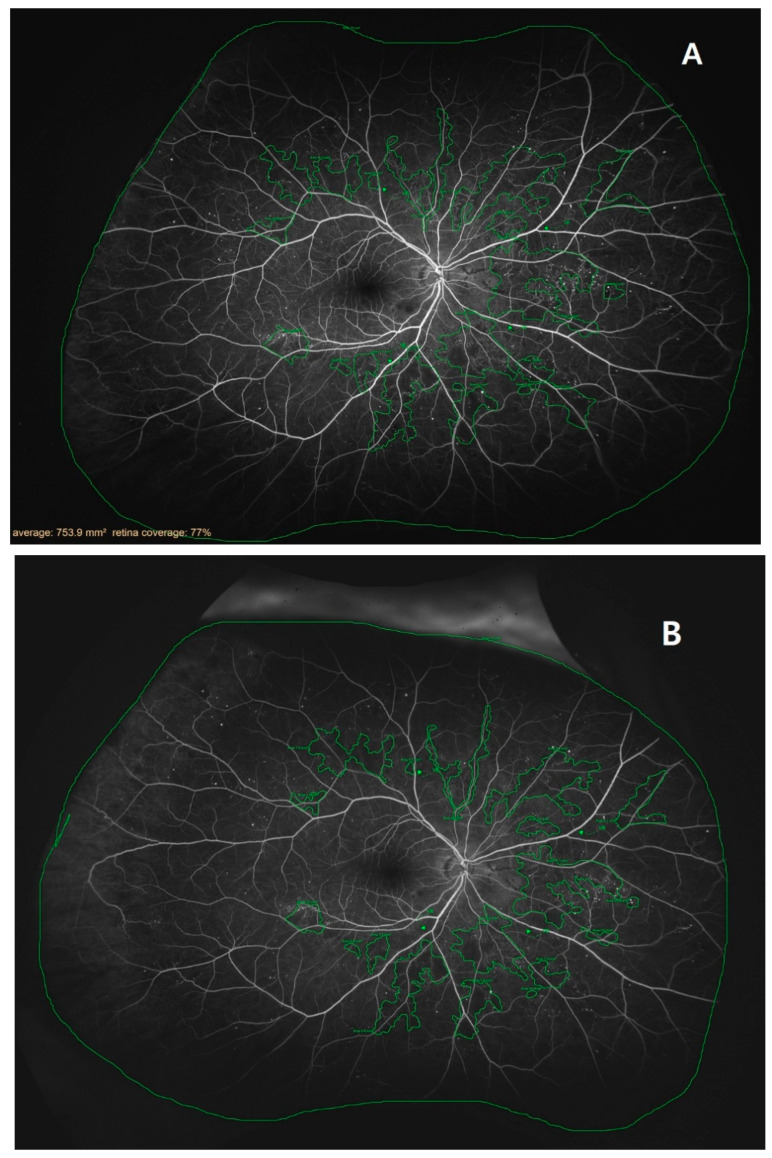
Example tracing images of retinas examined with ultrawide-field fluorescein angiography: (**A**) a retina 2 weeks before the treatment (V1); (**B**) a retina 8 ± 1 weeks after the first treatment (V9).

**Figure 2 medicina-59-00178-f002:**
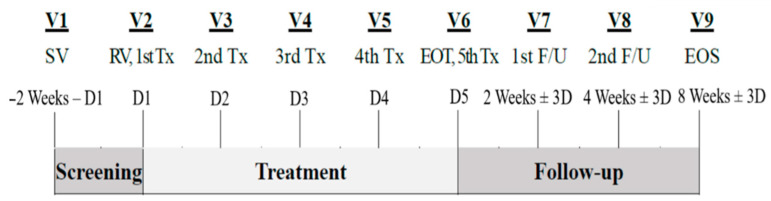
Study design: D, day; EOS, end of study; EOT, end of treatment; F/U, follow-up.

**Figure 3 medicina-59-00178-f003:**
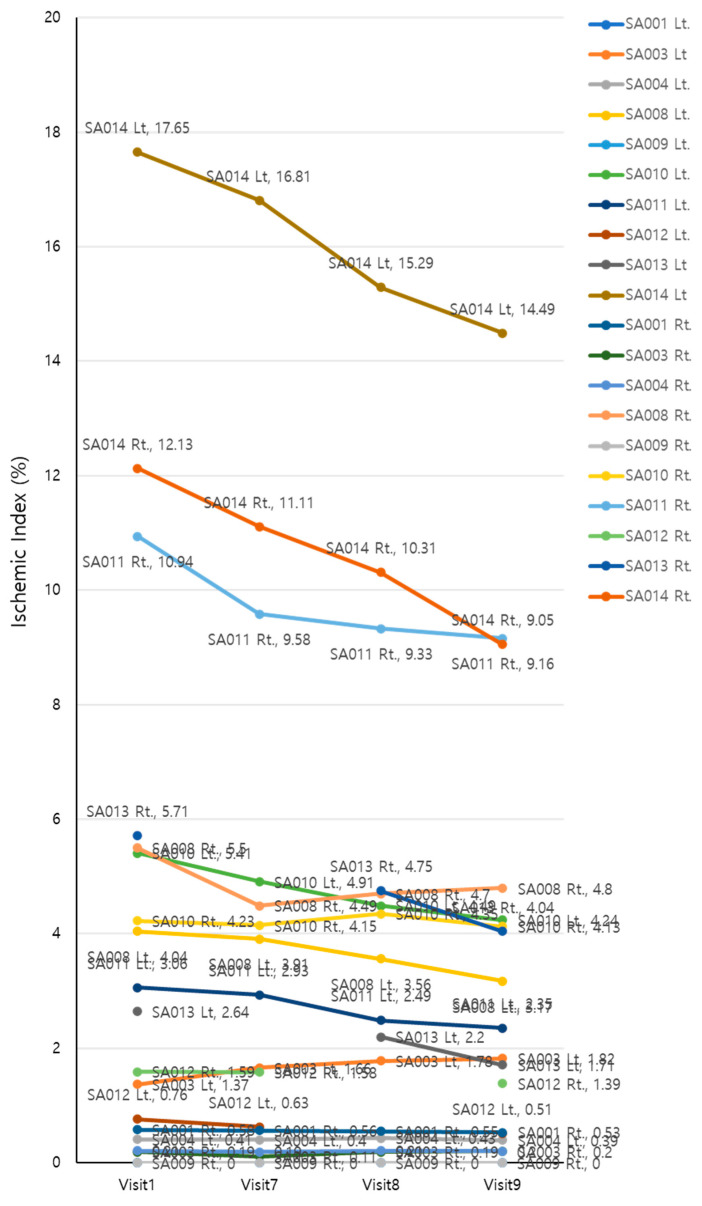
Changes in ischemic indexes of the participants during the study period.

**Figure 4 medicina-59-00178-f004:**
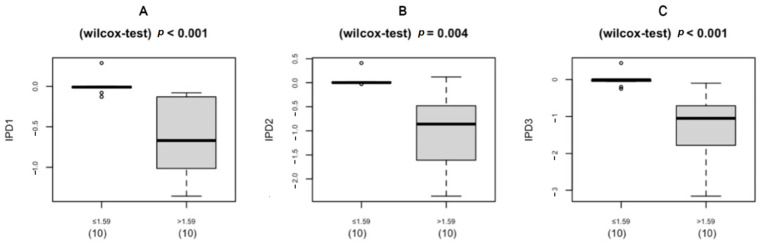
Boxplot of the IPD distribution for eyes below (left) and above (right) the cutoff of 1.59% at baseline (V1): (**A**) IPD1 (V1 vs. V7); (**B**) IPD2 (V1 vs. V8); (**C**) IPD3 (V1 vs. V9); IPD, ischemic percent difference; V, visit.

**Figure 5 medicina-59-00178-f005:**
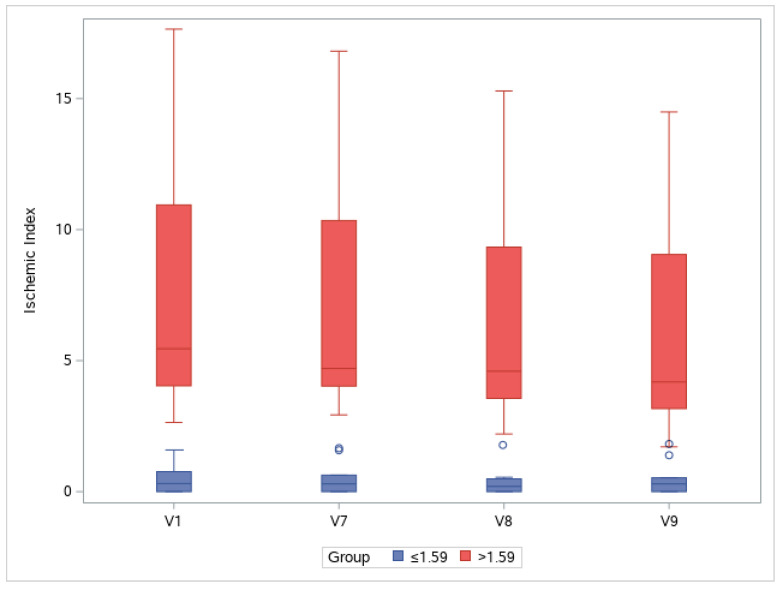
Changes in ischemic indexes throughout the study period. Groups are separated by a cutoff value of 1.59% at baseline (V1): V, visit.

**Figure 6 medicina-59-00178-f006:**
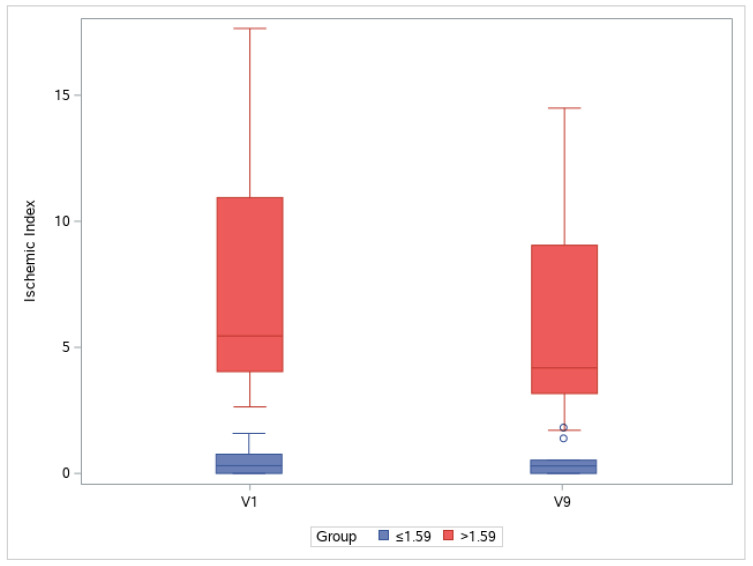
Changes in ischemic index between V1 and V9. Groups are separated by a cutoff value of 1.59% at baseline (V1): V, visit.

**Figure 7 medicina-59-00178-f007:**
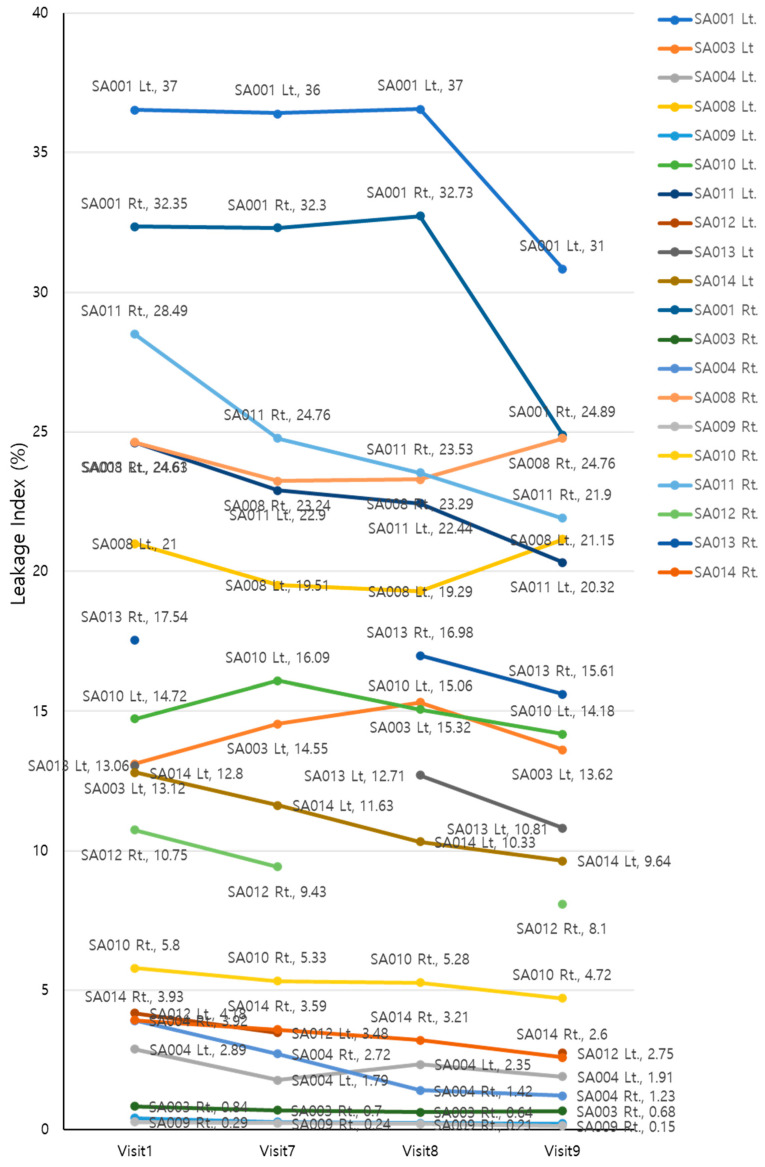
Changes in leakage indexes of the participants throughout the study period.

**Figure 8 medicina-59-00178-f008:**
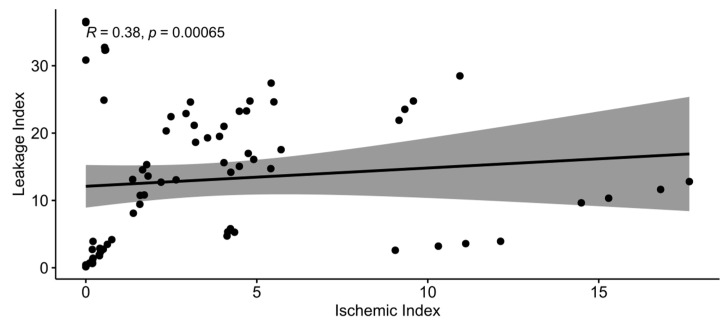
Spearman’s correlation between ischemic and leakage indexes.

**Table 1 medicina-59-00178-t001:** Patient demographics and main clinical findings.

Patients/Eyes	10/20
Age	61.50 ± 0.17
Gender (Male/Female)	8/2
Total Ischemic Retina	793.00 ± 20.03
Nonperfused Retina	29.78 ± 36.33
Ischemic Index	2.12 ± 4.78
Total Leakage Retina	796 ± 21.62
Capillary Leakage	108.65 ± 90.26
Leakage Index	13.59 ± 11.21

## Data Availability

The data presented in this study are available on request from the corresponding author.

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
