# Peer review of "A Nonrandomized Phase 2 Trial of EG-Mirotin, a Novel, First-in-Class, Subcutaneously Deliverable Peptide Drug for Nonproliferative Diabetic Retinopathy"

_medicina, 2023, doi:10.3390/medicina59010178_

Round 1
Reviewer 1 Report
Kindly remove line 37 from abstract section.
Please recheck the whole manuscript for typos and other errors, like line 46 there is no space between 2045 and reference 1.
Kindly provide the gap in the published work in the introduction section.
What is the novelty of your work?
How only 10 patients were recruited? How did you calculate the sample size?
Kindly mention in the limitation paragraph the small sample size recruitment for this study.
Author Response
Dear Reviewer/Doctor:
Season’s Greetings!
It is indeed my great pleasure to be reviewed by you for our article titled “A nonrandomized phase 2 trial of EG-Mirotin, a novel, first-in-class, subcutaneously delivered peptide drug for non-proliferative diabetic retinopathy.” I really appreciate it.
Kindly remove line 37 from abstract section.
Yes, we removed line 37
Please recheck the whole manuscript for typos and other errors, like line 46 there is no space between 2045 and reference 1.
Typographical errors are corrected
Kindly provide the gap in the published work in the introduction section.
The gap statement is added to the introduction section
What is the novelty of your work?
To date, ophthalmic treatments have progressed mainly for macular edema and neovascularization in diabetic retinopathy. However, it seems that no one has approached our treatment method as a treatment for capillary closure, which is the stage before macular edema and neovascularization. Unfortunately, there is still no effective treatment for nonproliferative diabetic retinopathy. The conventional treatment method was drug administration by intraocular injection, but EG-Mirotin is a subcutaneous injection method that has not been adopted in ophthalmology so far.
How only 10 patients were recruited? How did you calculate the sample size?
Since this study is IIa, as you know, to see the effects of drugs, the number of personnel entering the study met the minimum requirement of 10 people. It was decided after long discussions with the South Korean National Research Projects Officer.
Kindly mention in the limitation paragraph the small sample size recruitment for this study.
The small sample size was mentioned as a limitation in the discussion.
Again, thank you very much for your professional review!
Best wishes,
Sung Jin
Reviewer 2 Report
The paper is a good concept but the sample size is very less (8 men and 2 women). Sufficient samples are not there to test the Motrin peptide drug with NPDR.
Also, the samples are not sufficient for Type 1 and Type 2 diabetes. Both play different mechanisms, the study needs to be considered separately with sufficient samples and proper statistical methods need to be followed.
Also, hormonal changes also influence glucose metabolism so the gender difference study also needs to be considered separately for Type 1 and Type 2 diabetes.
Any other methods have been used for confirming the retina got affected like inflammation or layers of the retina and seen through either OCT or ERG.
Author Response
Dear Reviewer/Doctor:
Season’s Greetings!
It is indeed my great pleasure to be reviewed by you for our article titled “A nonrandomized phase 2 trial of EG-Mirotin, a novel, first-in-class, subcutaneously delivered peptide drug for non-proliferative diabetic retinopathy.” I really appreciate it.
I would like to respond to your opinions below:
The paper is a good concept but the sample size is very less (8 men and 2 women). Sufficient samples are not there to test the Motrin peptide drug with NPDR.
Since this study is IIa, as you know, to see the effects of drugs, the number of personnel entering the study met the minimum requirement of 10 people. It was decided after long discussions with the South Korean National Research Projects Officer.
Also, the samples are not sufficient for Type 1 and Type 2 diabetes. Both play different mechanisms, the study needs to be considered separately with sufficient samples and proper statistical methods need to be followed.
Thank you very much for your comment. After some investigation, we noticed that this study was only done for patients with type II DM
Also, hormonal changes also influence glucose metabolism so the gender difference study also needs to be considered separately for Type 1 and Type 2 diabetes.
It was addressed in the discussion section:
The small number of subjects in this trial limits our ability to precisely define safety and efficacy. In the IIb clinical trial that we are planning for the future, we believe that between 100 and 300 participants should be enrolled, which is a generally accepted number of participants.
Any other methods have been used for confirming the retina got affected like inflammation or layers of the retina and seen through either OCT or ERG.
Optical coherence tomography (OCT) and electroretinography (ERG) were not performed in this study as our trial focused on the early stage of diabetic retinopathy.
Again, thank you very much for your professional review!
Best wishes,
Sung Jin
Round 2
Reviewer 2 Report
Accept